# Effects of Growth Hormone Receptor Ablation in Corticotropin-Releasing Hormone Cells

**DOI:** 10.3390/ijms22189908

**Published:** 2021-09-14

**Authors:** Willian O. dos Santos, Daniela O. Gusmao, Frederick Wasinski, Edward O. List, John J. Kopchick, Jose Donato

**Affiliations:** 1Departamento de Fisiologia e Biofisica, Instituto de Ciencias Biomedicas, Universidade de Sao Paulo, Sao Paulo 05508-000, SP, Brazil; wswillbill@gmail.com (W.O.d.S.); daniogusmao@gmail.com (D.O.G.); frednefro@gmail.com (F.W.); 2Edison Biotechnology Institute and Heritage College of Osteopathic Medicine, Ohio University, Athens, OH 45701, USA; list@ohio.edu (E.O.L.); kopchick@ohio.edu (J.J.K.)

**Keywords:** adrenal axis, anxiety, corticosterone, GH, glucocorticoid, hypothalamus, metabolism, neuroendocrinology, paraventricular nucleus, stress

## Abstract

Corticotropin-releasing hormone (CRH) cells are the dominant neuronal population responsive to the growth hormone (GH) in the paraventricular nucleus of the hypothalamus (PVH). However, the physiological importance of GH receptor (GHR) signaling in CRH neurons is currently unknown. Thus, the main objective of the present study was to investigate the consequences of GHR ablation in CRH-expressing cells of male and female mice. GHR ablation in CRH cells did not cause significant changes in body weight, body composition, food intake, substrate oxidation, locomotor activity, glucose tolerance, insulin sensitivity, counterregulatory response to 2-deoxy-D-glucose and ghrelin-induced food intake. However, reduced energy expenditure was observed in female mice carrying GHR ablation in CRH cells. The absence of GHR in CRH cells did not affect anxiety, circadian glucocorticoid levels or restraint-stress-induced corticosterone secretion and activation of PVH neurons in both male and female mice. In summary, GHR ablation, specifically in CRH-expressing neurons, does not lead to major alterations in metabolism, hypothalamic–pituitary–adrenal axis, acute stress response or anxiety in mice. Considering the previous studies showing that central GHR signaling regulates homeostasis in situations of metabolic stress, future studies are still necessary to identify the potential physiological importance of GH action on CRH neurons.

## 1. Introduction

Growth hormone (GH) is secreted by the pituitary gland and stimulates protein synthesis and cell proliferation, as well as tissue and body growth [1,2,3]. GH also plays a key role in regulating several metabolic aspects, including glucose homeostasis and fat mobilization and oxidation [1,2,3]. Although less recognized, GH can also be considered a stress hormone. Several situations of metabolic stress are associated with increased GH secretion, which include hypoglycemia [4,5], physical exercise [6,7,8,9] and prolonged food deprivation [10,11,12]. The importance of GH secretion during situations of metabolic stress is not completely understood, but it may be associated with GH action promoting physiological adjustments in order to restore homeostasis [13]. In accordance with this idea, GH or GH receptor (GHR) deficiency increases the risk of hypoglycemia [14,15]. Mice that cannot produce the biologically active acetylated ghrelin exhibit profound hypoglycemia during food deprivation because of their incapacity to promote ghrelin-induced GH secretion and consequently increase hepatic glucose production [10,12]. Even in conditions that are not necessarily stressful, like puberty and pregnancy, there is a physiological increase in GH secretion [3,16]. However, these situations are characterized by higher energy requirements and changes in metabolism. Thus, GH secretion either during puberty or pregnancy may also produce important metabolic adjustments. Indeed, GH action can modulate insulin sensitivity, blood glucose levels and other metabolic aspects in juvenile humans and rodents [14,15,17]. Furthermore, the typical insulin resistance that emerges during puberty [18,19,20] is linked with the GH-insulin-like growth factor 1 axis [21,22]. Regarding pregnancy, a previous study has shown that GHR signaling is able to regulate food intake, body adiposity and insulin sensitivity in late pregnant mice [23]. Taken together, these studies show that GH secretion during situations of metabolic stress or higher energy requirements is necessary to produce appropriate physiological adjustments that help to cope with changes in homeostasis.

Increasing evidence indicates that the brain is an important target of GH in situations of metabolic stress [24]. Several brain nuclei involved in the regulation of metabolism and stress responses exhibit robust phosphorylation of the signal transducer and activator of transcription 5 (pSTAT5), a marker of GHR signaling, after a systemic GH injection [25,26]. These areas include the arcuate nucleus (ARH), ventromedial nucleus (VMH), paraventricular nucleus of the hypothalamus (PVH), bed nucleus of the stria terminalis (BNST), central nucleus of the amygdala (CEA) and locus coeruleus [25,26]. In the ARH, GH acts in several distinct neuronal populations, such as neurons that express agouti-related protein (AgRP) [11,23,27,28,29,30], proopiomelanocortin (POMC) [31], tyrosine hydroxylase [32,33] or choline acetyltransferase [34]. The role of GHR signaling in these neuronal populations is starting to be unraveled. For example, GHR expression in AgRP-expressing neurons is required to promote energy-saving adaptations during food restriction [11,27]. Additionally, mice carrying GHR ablation in AgRP neurons exhibit reduced blood glucose levels during food deprivation [11]. The absence of GHR signaling in POMC or AgRP neurons blunts glucoprivic hyperphagia [11,31]. GH action in VMH neurons regulates the counterregulatory response that recovers blood glucose levels from hypoglycemic conditions [5]. Another study has shown that GHR signaling in hypothalamic neurons regulates the adaptation capacity to aerobic exercise in a cell-specific manner [6]. Furthermore, GHR expression in dopaminergic neurons modulates stress-induced prolactin release [32]. Thus, central GH action promotes metabolic and neuroendocrine adaptations in different situations of metabolic stress.

Corticotropin-releasing hormone (CRH) is a key peptide involved in stress response [35,36]. PVH CRH-expressing neurons are particularly important in the stress response because part of these cells send axons to the median eminence to control the hypothalamic–pituitary–adrenal (HPA) axis. Thus, stress-induced glucocorticoid release depends on the activation of PVH CRH neurons. A recent study determined the neurochemical identity of GH-responsive neurons in the mouse PVH [37]. Although GH-induced pSTAT5 was found in different neuronal populations of the PVH, CRH-expressing neurons were the most abundant group of cells that were responsive to GH, representing 35% of all pSTAT5 cells in the PVH [37]. Importantly, 75% of the neuroendocrine and 74% of the non-neuroendocrine CRH neurons were responsive to GH in the PVH [37]. These findings suggest that CRH neurons possibly mediate GH action, particularly in stress responses. Therefore, the objective of the present study was to investigate the physiological importance of GHR expression in CRH neurons. For this purpose, we first determined the distribution of CRH neurons that are responsive to GH in the mouse brain. Subsequently, CRH-specific GHR knockout (KO) mice were generated and possible alterations in metabolism, ghrelin-induced food intake, basal and stress-induced corticosterone secretion, stress-induced activation of PVH neurons and anxiety were determined in male and female mice.

## 2. Results

### 2.1. Distribution of CRH Neurons That Are Responsive to GH

Initially, the distribution of CRH-expressing neurons that are responsive to GH was determined in the mouse brain. Thus, CRH reporter mice received a systemic GH injection and pSTAT5 staining was co-localized in CRH neurons of different brain nuclei. We observed some CRH neurons that were responsive to GH in the anterior division of the BNST (Figure 1a) and in the lateral hypothalamic area (LHA) (Figure 1b). A higher degree of co-localization was observed in the dorsomedial nucleus of the hypothalamus (DMH), particularly in its lateral part (Figure 1c). In addition, the majority of CRH neurons in the CEA were responsive to GH (Figure 1d). As previously described [37], approximately 70% of CRH neurons in the PVH co-localized with GH-induced pSTAT5 (Figure 1e). Taken together, these findings demonstrate that GH action can potentially modulate the activity of CRH neurons in different areas involved in stress response, especially PVH CRH neurons. To investigate the importance of GH action on CRH neurons, CRH-specific GHR KO mice were generated. To confirm the efficacy of the cell-specific GHR deletion, GH-induced pSTAT5 staining was co-localized in PVH CRH neurons of CRH GHR KO mice. Less than 8% of PVH CRH neurons exhibited GH-induced pSTAT5 in CRH GHR KO mice (Figure 1f,g), whereas pSTAT5 staining remained intact in non-CRH cells.

### 2.2. Reduced Energy Expenditure in CRH GHR KO Female Mice

CRH neurons are involved in the regulation of energy balance, food intake and other metabolic aspects [38,39,40,41,42]. To investigate whether GHR ablation in CRH neurons produce metabolic imbalances, possible changes in body weight, body composition, food intake and energy expenditure were determined in male and female mice. CRH GHR KO mice showed no differences in body weight, fat mass and lean body mass in both males (Figure 2a–d) and females (Figure 2i–l), compared to littermate control mice. Normal food intake was also observed in CRH GHR KO mice (Figure 2e,m). However, while CRH GHR KO male mice showed normal oxygen consumption (VO_2_) compared to control mice (Figure 2f), CRH GHR KO females exhibited reduced VO_2_ in the dark phase, light phase and over 24 h (Figure 2n). No significant differences in respiratory quotient and ambulatory activity were observed in male (Figure 2g,h) and female (Figure 2o,p) mice.

### 2.3. GHR Ablation in CRH Cells Does Not Affect Glucose Homeostasis and Counterregulatory Response

To evaluate possible changes in glucose homeostasis in mice carrying GHR ablation in CRH-expressing cells, glucose and insulin tolerance tests were performed. No differences in glucose tolerance were observed between control and CRH GHR KO mice either in males (Figure 3a) or females (Figure 3d). Likewise, GHR ablation in CRH neurons caused no alterations in insulin sensitivity in male (Figure 3b) and female (Figure 3e) mice. Glucocorticoid secretion is important to restore blood glucose levels during hypoglycemia [43,44]. Furthermore, GH action in specific brain sites regulates the counterregulatory response induced by 2-deoxy-D-glucose (2DG) injection [5]. Thus, the 2DG-induced counterregulatory response was analyzed in CRH GHR KO mice. However, CRH-specific GHR ablation did not affect the counterregulatory response induced by an acute 2DG injection in both male (Figure 3c) and female (Figure 3f) mice.

### 2.4. Ghrelin-Induced Food Intake Is Normal in CRH GHR KO Mice

The HPA axis is activated by ghrelin administration [45,46]. This occurs because the activity of PVH CRH neurons is modulated by ghrelin [45,47], although the mechanism of action is indirect [46,48]. Considering that both CRH and ghrelin regulate food intake in mice, and brain-specific GHR KO mice exhibit defects in the hyperphagia induced by systemic ghrelin injection [29], we determined whether ghrelin-induced food intake is intact in CRH GHR KO male mice (Figure 4). As expected [29], ghrelin injection induced a significant increase in food intake of control mice (*p* < 0.0001), as compared to saline-injected animals (Figure 4). However, no differences in ghrelin-induced food intake were observed between control and CRH GHR KO male mice (Figure 4).

### 2.5. GHR Ablation in CRH Cells Does Not Affect Circadian or Restraint Stress-Induced Corticosterone Secretion

Circulating corticosterone levels vary according to the time of the day. In this regard, circulating corticosterone concentration rises just before the onset of the active phase, whereas nadir corticosterone levels are observed during the sleep phase of the day. Considering that GH secretion also presents a circadian secretion pattern [49,50,51], we evaluated whether GHR ablation in CRH neurons may affect the circadian pattern of corticosterone secretion. We observed no differences between control and CRH GHR KO mice in plasma corticosterone levels measured at the morning or at the evening in both males (Figure 5a) and females (Figure 5d). Subsequently, restraint-stress-induced corticosterone secretion was evaluated. Restraint stress led to a robust increase in plasma corticosterone levels in male (*p* < 0.0001; Figure 5b,c) and female (*p* < 0.0001; Figure 5e,f) mice. However, GHR ablation in CRH neurons did not affect corticosterone response to restraint stress.

### 2.6. CRH GHR KO Mice Exhibit Normal Activation of PVH Neurons after Restraint Stress

PVH CRH neurons are robustly activated during acute stress [52]. Thus, we analyzed whether GHR expression in CRH neurons may alter the activation of PVH neurons after 120 min of restraint stress. Using Fos expression as a well-established marker of neuronal activation, we observed no significant differences between control and CRH GHR KO mice in the number of Fos-positive neurons in the PVH after restraint stress in either males (Figure 6a–c) or females (Figure 6d–f).

### 2.7. CRH GHR KO Mice Exhibit Normal Anxiety

CRH neurons in the PVH and other brain areas are key regulators of anxiety [36,53,54,55,56]. Therefore, possible changes in anxiety were evaluated in CRH GHR KO mice during the open field and elevated plus maze tests (Figure 7). During the open field test, control and CRH GHR KO male and female mice entered a similar number of times in the center (Figure 7a,e). Additionally, no difference in the time spent in the center of the open field was observed between control and CRH GHR KO mice (Figure 7b,f). Likewise, the number of entries and the time spent in the open arms during the elevated plus maze test were not different between the experimental mice in both males (Figure 7c,d) and females (Figure 7g,h). Of note, the distance traveled and the movement time in the open field and elevated plus maze tests were similar comparing control and CRH GHR KO mice (data not shown).

## 3. Discussion

In the present study, we investigated whether GHR expression in CRH cells is necessary to modulate different physiological aspects controlled by these neurons. Despite the wide responsiveness to GH in different CRH neuronal populations, GHR ablation caused no metabolic, neuroendocrine or behavioral alterations in mice, except for a reduced energy expenditure observed in CRH GHR KO female mice.

Our neuroanatomical findings not only confirm previous studies showing a high percentage of PVH CRH neurons responsive to GH [37], but they also demonstrate the presence of GH-induced pSTAT5 in CRH neurons of other brain areas. CRH neurons in the LHA are activated by dehydration/osmotic stimulation and they possibly regulate homeostatic and behavioral responses during this type of stress [57,58,59]. However, we did not test whether CRH GHR KO mice exhibit altered responses to dehydration. The distribution of CRH neurons in the lateral DMH showed here is in accordance with previous studies [52,60] and the in situ hybridization data available in the Allen Brain Atlas (https://mouse.brain-map.org/experiment/show/292, accessed on 10 August 2021). Nonetheless, the physiological importance of DMH CRH neurons remains unknown. CEA CRH neurons send descending inputs to the parabrachial nucleus [61,62]. These inhibitory projections are possibly involved in the modulation of pain perception and chronic pain [63]. Future studies are necessary to investigate whether the response to pain is altered in CRH GHR KO mice.

Given the high percentage of CRH neurons responsive to GH in the PVH, our study mainly focused on investigating the biological functions classically regulated by this neuronal population. PVH CRH neurons are the master regulators of the HPA axis and stress responses in general. However, CRH GHR KO mice exhibited a normal circadian pattern of glucocorticoid levels and no differences in acute restraint-stress-induced glucocorticoid secretion. In accordance with these findings, restraint-stress-induced activation of PVH neurons, evaluated via Fos expression, was similar between control and CRH GHR KO mice. Therefore, GH action in CRH neurons does not modulate the acute responses to stress. In contrast to our findings, GHR signaling in dopamine neurons regulates prolactin release during restraint stress [32]. GHR ablation in VMH neurons acutely impairs the counterregulatory response induced by 2DG or insulin [5], which is another form of metabolic stress. Although the normal phenotype of CRH GHR KO mice may simply indicate that GH action on CRH neurons is irrelevant in situations of stress, it is also possible that GHR signaling may modulate the stress response in a more complex way. In this sense, acute stress leads to complex changes in behavior that emerge just after the stimulus [36]. These alterations are modulated by PVH CRH neurons [36]. Therefore, we cannot rule out the possibility that GHR signaling may modulate behavioral responses or other physiological aspects that become apparent after the stress and were not evaluated in our study. It is also important to point out that we did not evaluate chronic or repeated stress. Of note, central GH action is involved in adaptive changes following prolonged stress [64,65]. Previous studies have shown that ghrelin levels rise during repeated stress and ghrelin stimulates GH expression in the amygdala [64,66]. GH overexpression in the amygdala leads to increased fear, which is a maladaptive response to chronic stress and is possibly involved in excessive fear memory formation associated with posttraumatic stress disorder [64,65]. Thus, although our study indicates that GHR ablation in CRH neurons does not affect acute stress response, future studies will be important to clarify whether central GH action recruits CRH neurons, particularly in the PVH or CEA, to favor the development of excessive fear memory formation during prolonged stress.

The CRH system plays a pivotal role regulating anxiety [36,53,54,55,56]. Anxiety is controlled by CRH neurons in the PVH [36,53,54], CEA [67] and BNST [68,69,70]. Possible changes in anxiety were determined in the present study through the open field and elevated plus maze tests, which are widely used for this purpose [71,72]. Briefly, increases in the time spent in the center of the open field and in the open arms or in the number of entries into these areas indicate anti-anxiety behavior [71,72]. Nonetheless, CRH GHR KO mice did not exhibit evidence of changes in anxiety, as compared to control mice. Since our study assessed changes in anxiety in naïve (non-stressed) mice, future studies should investigate the potential role of GH action in CRH neurons regulating anxiety after acute and chronic stress.

The importance of CRH neurons regulating feeding and metabolism is well established [38,39,40,41,42]. Similarly, central GHR signaling potentially regulates food intake [11,23,31,73], energy expenditure [11,27], body adiposity [23] and glucose homeostasis [11,23], particularly in certain situations, like food deprivation, pregnancy and after hypoglycemia. We did not observe significant alterations in metabolism, at least in mice consuming a regular rodent chow. The exception was a reduced energy expenditure exhibited by CRH GHR KO females. High-fat/high-caloric diet disturbs the activity of PVH CRH neurons, leading to decreased energy expenditure [74]. Intriguingly, it is unknown why the low energy expenditure was observed only in CRH GHR KO female mice. GH secretion also presents a well-known sex difference [49,50,51]. Additionally, CRH neurons exhibit sexually dimorphic neuronal responses to different forms of stress, such as social isolation or changes in prenatal maternal environment [75,76,77]. These gender-specific differences may be related to estrogen signaling in CRH neurons. PVH CRH neurons express estrogen receptor β [78], as well as the membrane-associated estrogen receptor, whose activation induces glutamatergic excitatory postsynaptic currents [79]. However, this change in energy expenditure was not followed by alterations in body weight, body adiposity or compensatory changes in food intake. Therefore, the biological meaning of the reduced VO_2_ exhibited by CRH GHR KO female mice remains uncertain.

The feeding response induced by ghrelin administration is blunted in GHR KO mice or in mice carrying GHR ablation in the entire nervous system [29,80]. Although ghrelin acts on AgRP neurons to increase food intake [48,81,82], GHR ablation in AgRP neurons does not affect the hyperphagia induced by ghrelin [29]. Considering that PVH CRH neurons are indirectly activated by ghrelin [45,46,47,48], we tested whether GHR expression in CRH neurons could influence the feeding response to ghrelin. However, CRH GHR KO mice exhibited a normal orexigenic response to systemic ghrelin injection, indicating that CRH neurons do not mediate the effects of GH on the feeding response to ghrelin. Importantly, ghrelin-induced food intake was evaluated only in male mice, whereas the remaining experiments were performed in both sexes. We used only male mice in this specific experiment because previous studies have indicated that females exhibit a lower orexigenic response to ghrelin [83].

In conclusion, our findings show that CRH neurons in different brain areas, particularly in the PVH, are highly responsive to GH. However, GHR ablation in CRH neurons caused no significant alterations in metabolism, HPA axis, acute stress response or anxiety in mice. However, additional challenges in CRH GHR KO mice, including prolonged stress exposure, other forms of stress (dehydration or pain) and high-fat diet, still need to be evaluated in order to determine the potential role of GHR signaling in CRH neurons. Additionally, the evaluation of short- and long-term consequences of an acute stress in CRH GHR KO mice may provide valuable insights regarding the potential role of central GHR signaling regulating stress response.

## 4. Materials and Methods

### 4.1. Mice

CRH reporter mice were produced by crossing CRH-Cre mice (The Jackson Laboratory; RRID: IMSR_JAX:012704) with a Cre-dependent tdTomato reporter mouse (The Jackson Laboratory; RRID: IMSR_JAX:007909). CRH-specific GHR ablation was achieved by breeding mice carrying loxP-flanked *Ghr* alleles [84] with CRH-Cre mice. CRH GHR KO mice were homozygous for the loxP-flanked *Ghr* alleles and carried one copy of the Cre transgene, whereas control mice were littermates, homozygous for the loxP-flanked *Ghr* alleles without carrying the Cre gene. The mice used in the experiments were from the C57BL/6 background and their mutations were confirmed by genotyping the tail tip collected during weaning. The experiments were approved by the Ethics Committee on the Use of Animals of the Institute of Biomedical Sciences at the University of São Paulo (protocol number: 73/2017) and were performed according to the ethical guidelines adopted by the Brazilian College of Animal Experimentation.

### 4.2. Detection of GH Responsive neurons

Mice received an intraperitoneal (i.p.) injection of porcine pituitary GH [20 µg/g of body weight from Dr. A.F. Parlow, National Institute of Diabetes and Digestive and Kidney Diseases—National Hormone and Pituitary Program (NIDDK-NHPP)] and were perfused 90 min later. Mice were deeply anesthetized with isoflurane and perfused transcardially with saline, followed by a 10% buffered formalin solution. Brains were collected and post-fixed in the same fixative for 45 min and cryoprotected overnight at 4 °C in 0.1 M phosphate-buffered saline (PBS) containing 20% sucrose. Brains were cut in 30 µm thick sections using a freezing microtome. Brain slices were rinsed in 0.02 M potassium PBS, pH 7.4 (KPBS), followed by pretreatment in a water solution containing 1% hydrogen peroxide and 1% sodium hydroxide for 20 min. After rinsing in KPBS, sections were incubated in 0.3% glycine and 0.03% lauryl sulfate for 10 min each. Next, slices were blocked in 3% normal donkey serum for 1 h, followed by incubation in anti-pSTAT5^Tyr694^ antibody (1:1000; Cell Signaling Technology; Danvers, MA, USA, Cat# 9351; RRID: AB_2315225) for 40 h. Subsequently, sections were rinsed in KPBS and incubated for 90 min in Alexa Fluor^488^-conjugated secondary antibody (1:500, Jackson ImmunoResearch, West Grove, PA, USA). The visualization of tdTomato-expressing neurons does not require any staining. After rinses in KPBS, sections were mounted onto gelatin-coated slides and the slides were covered with Fluoromount G mounting medium (Electron Microscopic Sciences; Hatfield, PA, USA).

### 4.3. Metabolic Measurements

The body weight of male and female mice were determined weekly from weaning up to 5 months of life. After that, body fat mass and lean body mass were determined by time-domain nuclear magnetic resonance using the LF50 body composition mice analyzer (Bruker, Germany). Subsequently, food intake was measured for approximately 4 consecutive days in single-housed mice. VO_2_, CO_2_ production, ambulatory activity (by infrared sensors) and respiratory quotient (CO_2_ production/VO_2_) were determined using the Oxymax/Comprehensive Lab Animal Monitoring System (Columbus Instruments, Athens, OH, USA) for 5–7 days. For the glucose, insulin and 2DG tolerance tests, food was removed from the cage for 4 h. After evaluating basal glucose levels (time 0), mice received i.p. injections of 2 g glucose/kg, 1 IU insulin/kg or 0.5 g 2DG/kg (Sigma, St. Louis, MO, USA), respectively, followed by serial determinations of blood glucose levels using a glucose meter through samples collected from the tail tip.

### 4.4. Evaluation of Ghrelin-Induced Food Intake

Ghrelin-induced food intake was analyzed in 2–3 month-old male mice that received an i.p. injection of saline or ghrelin (0.2 µg/g b.w., Global Peptide, cat #C-et-004). The amount of food ingested was determined 30, 60 and 120 min after the injections that were performed in ad libitum fed mice, approximately three hours after lights on.

### 4.5. Evaluation of Circadian and Restraint-Stress-Induced Corticosterone Secretion

The circadian pattern of corticosterone secretion was evaluated in male and female mice in the morning (09:00 a.m.; lights on at 08:00 a.m.; 12 h light/dark) and in the evening (19:00). Plasma samples were obtained after centrifuging blood collected from the tail vein using heparinized capillaries (75 mm length and 1.5 mm diameter). To determine restraint-stress-induced corticosterone secretion, mice at 09:00 a.m. were physically restrained for 120 min in 50 mL polypropylene centrifuge tubes, containing holes of 0.4 cm diameter to breathe, as previously described [32]. A small cut in the tail tip was performed to allow blood collection using heparinized capillaries. The blood was collected immediately following the onset of stress (time 0) and after 15, 30, 45, 60, 90 and 120 min of restraint stress. A commercially available enzyme-linked immunosorbent assay kit was used to measure plasma corticosterone levels (Arbor Assays; Ann Arbor, MI, USA; cat#K014-H5).

### 4.6. Analysis of Stress-Induced Fos Expression

After 120 min of restraint stress, control and CRH GHR KO male and female mice were perfused and the brains processed as described in the reaction to detect GH responsive cells. To label Fos protein, brain sections were rinsed in KPBS, pretreated in 3% normal donkey serum for 1 h and incubated in anti-Fos antibody (1:10,000, Ab5, Millipore, Temecula, CA, USA; RRID: AB_2314043) for 40 h. Subsequently, sections were rinsed in KPBS, incubated for 90 min in Alexa Fluor^488^-conjugated secondary antibody (1:500, Jackson ImmunoResearch) and mounted onto gelatin-coated slides.

### 4.7. Open Field and Elevated Plus Maze Tests

For the open field test, mice were placed in an open field arena (40 cm [w] × 40 cm [d] × 30 cm [h]) for 5 min. The number of entries and the time spent in the center of the arena were determined. After two days, mice were tested in the elevated plus maze test. We used a maze that was elevated 40 cm from the ground and was composed of two opposite open arms measuring 35 × 5 cm, crossed by two closed arms of similar dimensions. Each animal was placed in the center of the apparatus and the number of entries and the time spent in the open arms were measured for 5 min. In both tests, the animal’s position and movements were recorded and analyzed by the ANY-maze software and the experiments were performed between 12:00 and 04:00 p.m. After each test, the apparatuses were cleaned with 70% ethanol and air-dried before a new trial.

### 4.8. Statistical Analysis

Differences between groups were analyzed by unpaired two-tailed Student’s *t*-test. Changes along time were determined by repeated measures two-way ANOVA. Statistical analyses were performed using the Prism software (GraphPad). *p* values < 0.05 were considered statistically significant. All results were expressed as mean ± standard error of the mean.

## Figures and Tables

**Figure 1 ijms-22-09908-f001:**
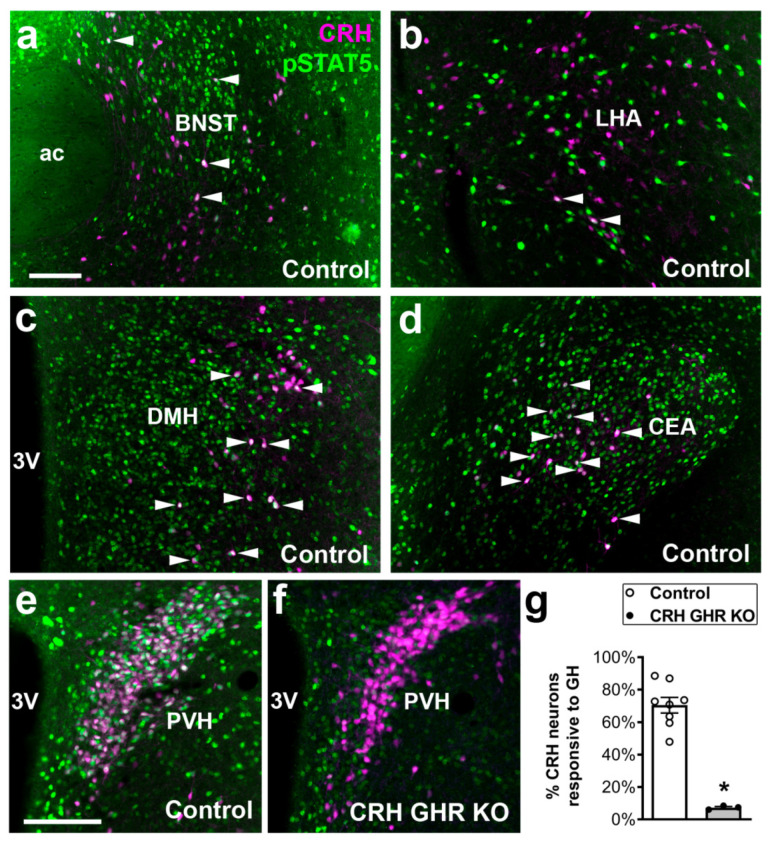
Distribution of CRH neurons that are responsive to GH. (**a**–**e**) Epifluorescence photomicrographs illustrating the co-localization between pSTAT5 (green nuclear staining) and CRH (magenta cytoplasmic marking) in GH-injected control mice. Arrowheads indicate examples of double-labeled neurons (appear as white). (**f**) Significant reduction of co-localizations between pSTAT5 and CRH in GH-injected CRH GHR KO mice. (**g**) Quantification of the percentage of PVH CRH neurons that are responsive to GH in control (*n* = 8) and CRH GHR KO mice (*n* = 3). * *p* < 0.0001. Abbreviations: 3V, third ventricle; ac, anterior commissure. Scale bars: (**a**) = 100 µm; (**e**) = 50 µm.

**Figure 2 ijms-22-09908-f002:**
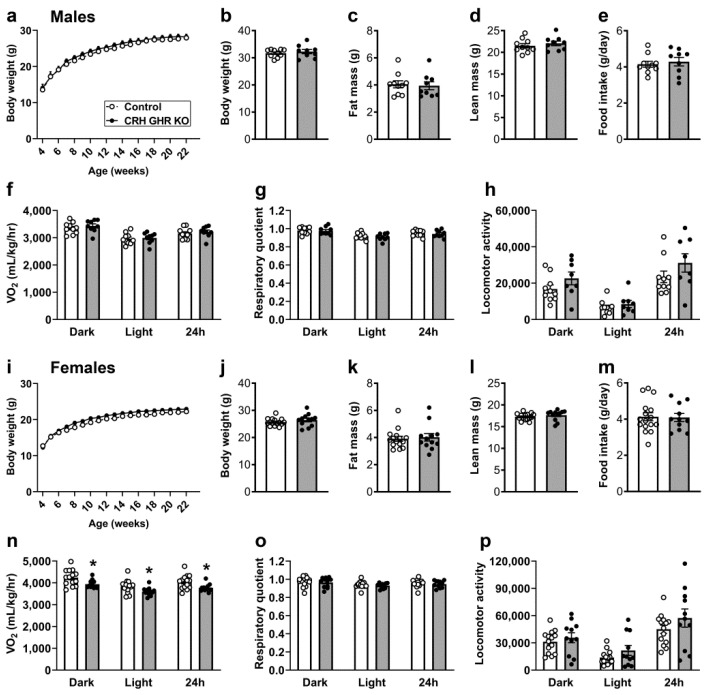
Reduced energy expenditure in CRH GHR KO female mice. (**a**) Changes in body weight along time in control (*n* = 10) and CRH GHR KO (*n* = 9) male mice. (**b**) Body weight. (**c**) Body fat mass. (**d**) Lean body mass. (**e**) Daily food intake. (**f**) Oxygen consumption (VO_2_). (**g**) Respiratory quotient. (**h**) Ambulatory activity. (**i**) Changes in body weight along time in control (*n* = 15–16) and CRH GHR KO (*n* = 11–12) female mice. (**j**) Body weight. (**k**) Body fat mass. (**l**) Lean body mass. (**m**) Daily food intake. (**n**) VO_2_. (**o**) Respiratory quotient. (**p**) Ambulatory activity. * *p* < 0.05.

**Figure 3 ijms-22-09908-f003:**
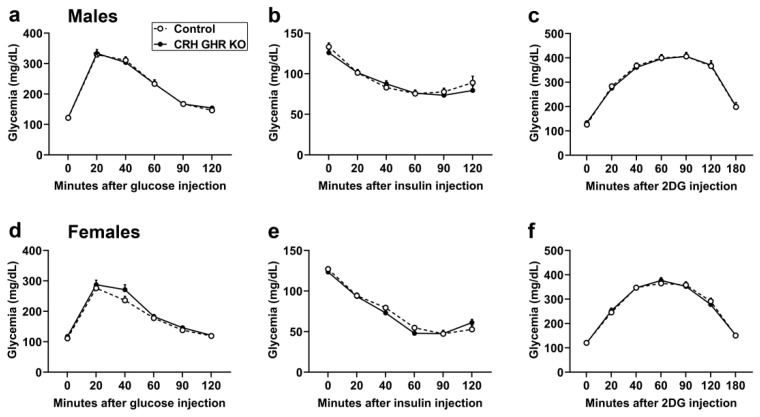
GHR ablation in CRH cells does not affect glucose homeostasis and counterregulatory response. (**a**–**c**) Glucose tolerance test (**a**), insulin tolerance test (**b**) and counterregulatory response (**c**) to 2-deoxy-D-glucose (2DG) in control (*n* = 10) and CRH GHR KO (*n* = 9) male mice. (**d**–**f**) Glucose tolerance test (**d**), insulin tolerance test (**e**) and counterregulatory response to 2DG (**f**) in control (*n* = 16) and CRH GHR KO (*n* = 11) female mice.

**Figure 4 ijms-22-09908-f004:**
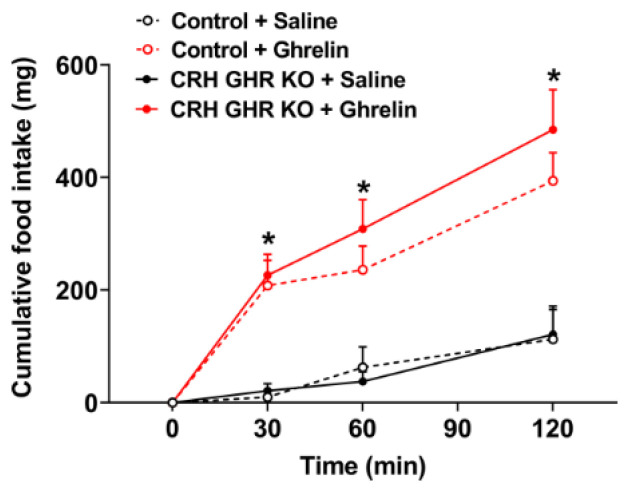
Ghrelin-induced food intake is normal in CRH GHR KO mice. Cumulative food intake in control (*n* = 5) and CRH GHR KO (*n* = 8) male mice that received saline or ghrelin injection. * Ghrelin-injected groups are significantly different compared to saline-injected groups (*p* < 0.05).

**Figure 5 ijms-22-09908-f005:**
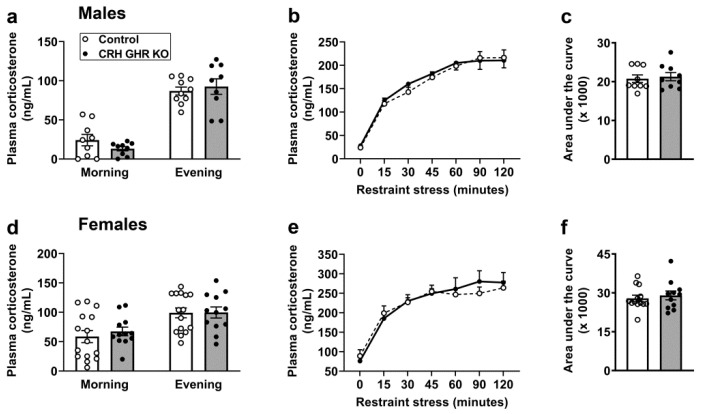
GHR ablation in CRH cells does not affect circadian or restraint-stress-induced corticosterone secretion. (**a**) Plasma corticosterone levels in the morning (09:00 a.m.; lights on at 08:00 a.m.) and in the evening (19:00) in control (*n* = 9–10) and CRH GHR KO (*n* = 9) male mice. (**b**,**c**) Changes in plasma corticosterone levels during restraint stress in control (*n* = 9) and CRH GHR KO (*n* = 9) male mice. (**d**) Plasma corticosterone levels in the morning and in the evening in control (*n* = 15) and CRH GHR KO (*n* = 12) female mice. (**e**,**f**) Changes in plasma corticosterone levels during restraint stress in control (*n* = 13) and CRH GHR KO (*n* = 11) female mice.

**Figure 6 ijms-22-09908-f006:**
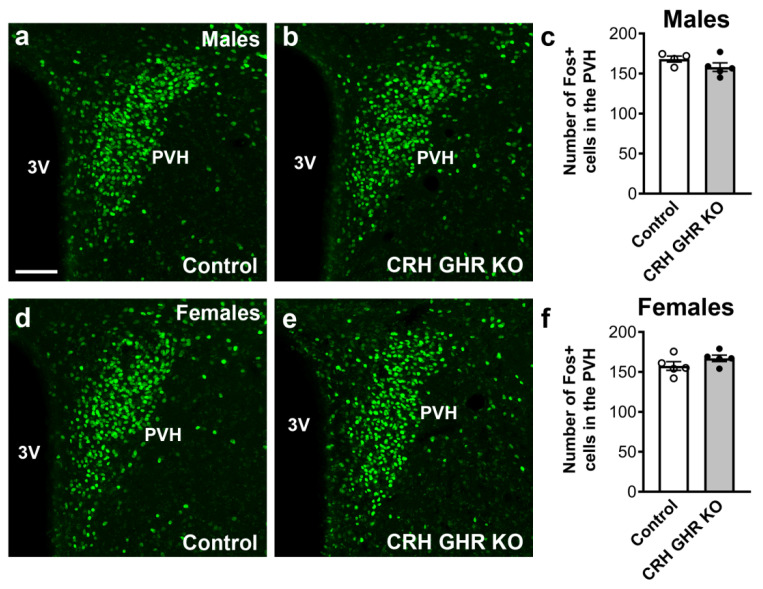
CRH GHR KO mice exhibit normal activation of PVH neurons after restraint stress. (**a**–**c**) Restraint-stress induced Fos expression in the PVH of control (white circles; *n* = 4) and CRH GHR KO (black circles; *n* = 5) male mice. (**d**–**f**) Restraint-stress induced Fos expression in the PVH of control (*n* = 5) and CRH GHR KO (*n* = 5) female mice. Abbreviation: 3V, third ventricle. Scale bar = 100 µm.

**Figure 7 ijms-22-09908-f007:**
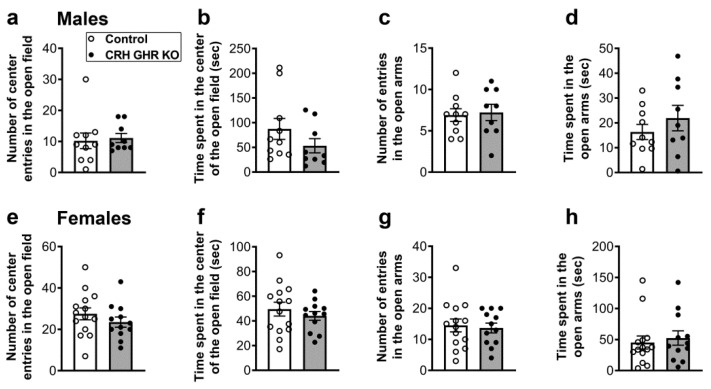
CRH GHR KO mice exhibit normal anxiety. (**a**,**b**) Number of entries (**a**) and time spent (**b**) in the center of the open field in control (*n* = 10) and CRH GHR KO (*n* = 9) male mice. (**c**,**d**) Number of entries (**c**) and time spent (**d**) in the open arms during the elevated plus maze test in control (*n* = 10) and CRH GHR KO (*n* = 9) male mice. (**e**,**f**) Number of entries (**e**) and time spent (**f**) in the center of the open field in control (*n* = 14) and CRH GHR KO (*n* = 12) female mice. (**g**,**h**) Number of entries (**g**) and time spent (**h**) in the open arms during the elevated plus maze test in control (*n* = 14) and CRH GHR KO (*n* = 12) female mice.

## Data Availability

The data that support the findings of this study are available from the corresponding author upon reasonable request.

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
