# Peer review of "Effects of Growth Hormone Receptor Ablation in Corticotropin-Releasing Hormone Cells"

_ijms, 2021, doi:10.3390/ijms22189908_

Round 1

Reviewer 1 Report

The paper presents predominantly negative results. It is not clear whether this is due to GHR KO leading to additional changes at metabolic level

Author Response

We appreciate the reviewer’s comments and the time spent evaluating our manuscript. Our findings show that CRH neurons are highly responsive to GH in different brain areas. However, GHR ablation in CRH neurons caused no significant alterations in metabolism, HPA axis, acute stress response or anxiety in mice.  Although these results are predominantly negative, they are important because CRH neurons represent the most abundant neuronal population responsive to GH in the paraventricular nucleus of the hypothalamus, a key site that controls metabolism, endocrine axes, autonomic function, anxiety and motivated behaviors. We extensively discussed the possible explanations for these negative results. Very briefly, although GHR ablation in CRH neurons did not lead to changes in the basal state or in acute stress response, we suggested that future studies should investigate additional challenges in this mouse model, including prolonged/repeated stress exposure, other forms of stress (dehydration or pain) and high-fat diet intake. Therefore, our study represents an important starting point to show that CRH neurons are targets of GH and that although some initial analyzes showed negative results, it is worth investigating in more detail the potential importance of GHR expression in these cells.

Reviewer 2 Report

This a very interesting manuscript, although I think that further studies will reveal the real significance of the presence of GHR in CRH neurons. For instance, do you think that increasing the time at which the studies have been performed (one, two weeks) would reveal the meaning of the detection of GHR in GHR neurons? It is there an explanation for the differences between make and female GHR KO mice regarding VO2, such as estrogens or GnRH? Could be useful to measure plasma levels of catecholamines? Many questions arise from your data, but I am sure that you will find an answer to them. Perhaps the Introduction is too long and it could be shortened, but I don't have any problem with it. 

Author Response

We thank the reviewer for their constructive comments. In the present paper, we discussed that our study represents an important starting point to show that CRH neurons are targets of GH, and that although some initial analyzes showed negative results, future studies could investigate the potential importance of GHR expression in these cells through additional challenges, including prolonged/repeated stress exposure, other forms of stress (dehydration or pain) and high-fat diet intake. Thus, rather than increasing the duration of the study or specific experiments, it may be more promising to test different challenges, especially chronic stress conditions. We extensively discussed these possibilities in the manuscript. We do not have an explanation for the difference in energy expenditure between male and female CRH GHR KO mice, but it may be related to estrogen signaling in CRH neurons. To discuss this possibility, a phrase was added in the revised manuscript (page 10):

“High-fat/high-caloric diet disturbs the activity of PVH CRH neurons leading to decreased energy expenditure [79]. Intriguingly, it is unknown why the low energy expenditure was observed only in CRH GHR KO female mice. GH secretion also presents a well-known sex difference [1-3]. Additionally, CRH neurons exhibit sexually dimorphic neuronal responses to different forms of stress, such as social isolation or changes in prenatal maternal environment [80-82]. These gender-specific differences may be related to estrogen signaling in CRH neurons. PVH CRH neurons express estrogen receptor β [83], as well as membrane-associated estrogen receptor, whose activation induces glutamatergic excitatory postsynaptic currents [84]. However, this change in energy expenditure was not followed by alterations in body weight…”

Catecholamine levels (activation of the sympathetic nervous system) could be another exciting indicator of a stress response. Unfortunately, we do not have material collected to assess adrenaline or noradrenaline. However, given the fact that several markers of acute stress response were normal in CRH GHR KO mice (e.g., corticosterone levels, activation of PVH neurons or anxiety), perhaps it would be more interesting to analyze the levels of catecholamines after repeated stress. Thus, we will consider this suggestion in future studies. Finally, we reduced the Introduction section in 134 words (from 890 to 756 words; 15% reduction).